Cross-cultural; Adaptation; Discrimination; Assessment tools; Mental illness

**Corresponding author:**
Santosh Loganathan
Email: dr.santosh32@gmail.com

# Cultural adaptation process of six stigma assessment scales among Kannada speaking population in South India

Harshitha H Annajigowda[1] , Gurucharan Bhaskar Mendon[2] , Anish V. Cherian[2],
Syed Shabab Wahid[3] , Brandon A. Kohrt[4] , Nicolas Rüesch[5],
Sara Evans-Lacko[6] , Elaine Brohan[7], Claire Henderson[8] ,
Graham Thornicroft[9] and Santosh Loganathan[10]

[1]Department of Psychiatry, National Institute of Mental Health and Neurosciences, Bengaluru, India; [2]Department of Psychiatry Social Work, National Institute of Mental Health and Neurosciences, Bengaluru, India; [3]Department of Global Health, School of Health, Georgetown University, Washington, DC, USA; [4]Center for Global Mental Health Equity, Department of Psychiatry, George Washington University, Washington DC, USA; [5]Department of Psychiatry II, Ulm University and BKH Günzburg, Germany; [6]Health Service and Population Research Department, Institute of Psychiatry, Psychology and Neuroscience, King's College London, London UK; [7]Centre for Global Mental Health, Health Service and Population Research Department, Institute of Psychiatry, Psychology and Neuroscience, King's College London, London, UK; [8]Public Mental Health, Health Services and Population Research Department P029, David Goldberg Centre, King's College London Institute of Psychiatry, Psychology and Neuroscience, De Crespigny Park, London, UK; [9]Centre for Global Mental Health and Centre for Implementation Science, Institute of Psychiatry, Psychology and Neuroscience, King's College London, London, UK and [10]Department of Psychiatry, National Institute of Mental Health and Neurosciences (NIMHANS), Bengaluru, India

## Abstract

For several years stigma researchers in India have relied on Western instruments or semi-structured stigma scales in their studies. However, these scales have not been rigorously translated and adapted to the local cultural framework. In the current study, we describe the cultural adaptation of six stigma scales with the purpose of using it in the native language (Kannada) based on translation steps of forward translation, expert review and synthesis, cultural equivalence, back translation and cognitive interview processes.

Several items were modified in the target language at each stage of the cultural adaptation process as mentioned in the above steps across all scales. Cultural explanations for the same have been provided. Concepts such as "community forest" and "baby sitting" was replaced with equivalent native synonyms. We introduced native cultural and family values such as "joint family system" and modified the item of housing concept in one of the tools. The concept of "privacy" in the Indian rural context was observed to be familial than individual-based and modification of corresponding items according to the native context of "privacy". Finally, items from each scale were modified but retained without affecting the meaning and the core construct.

## Impact Statement

The advantage of using standardized validated instruments is that the tool measures what it is supposed to measure and could be guaranteed consistency across different sociodemographic areas and across raters. For several years stigma researchers in India have either relied on Western instruments or semi-structured stigma scales in their studies. Several of the semi-structured stigma scales that were used lack the rigorous standardization that is required for any scale to be used consistently and repeatedly. Moreover, stigma and discrimination are also commonly experienced in the context of caste, gender and poverty in the Indian socio-cultural context. As a result, the scales measuring stigma related to mental illness need to consider these socio-cultural contexts. In this cross-cultural adaptation process, we have incorporated the vital steps described in the literature that would best eliminate the possible biases at the source. A five-step process of cross-cultural adaptation will be elucidated in this study in the process of cultural adaptation of six standardized stigma scales to one of the native Indian languages Kannada which is spoken in the South Indian state of Karnataka. This study was conducted in the rural part of Karnataka, Ramanagaram district. This study explains the importance of content validity and cultural equivalence in the adaptation process. It also introduces cultural and linguistic aspects of the target language and incorporation of same during the process of adaptation. We hope that the cultural adaptation process described in this study is useful for other researchers wanting to decode and adapt these scales to other languages in India and possibly other low- and middle-income countries.

## Introduction

The prevalence of psychiatric disorders in India as per the most recent National Mental Health Survey (NMHS), is 10.6% with an overall treatment gap of 83% (Gururaj *et al.* 2016). Several demand-side barriers like lack of awareness, stigma, financial difficulties, distance from the clinic, doubts about the efficacy of care and supply-side barriers, such as lack of hygiene, long waits in government hospitals and high service costs delay the access to mental healthcare services contributing to this treatment gap (Almanzar *et al.* 2014; Saraceno *et al.* 2007). There is a growing need for anti-stigma interventions, and the evaluation of these interventions requires an ecologically valid and culturally sensitive set of tools to assess the outcomes. A systematic review suggests that three-fourths of studies used adaptations of existing Western stigma measures to different cultural settings emphasizing the importance of conducting both quantitative and qualitative research to develop culture-specific stigma measures in everyday practice (Yang *et al.* 2014).

Psychometric properties like test–retest reliability and validity are important, but they may not necessarily identify the item bias and the inadequacy in equivalence between the original and the translated scale may make the tool a biased one (S M Yasir Arafat 2016). The content validity is best assessed by means of expert evaluation and judgment of the target population, which refers to the population for which these scales will be used (Boateng *et al.* 2018). The use of standardized and validated research tools is important for universal applicability and comparisons across countries and different timelines (Gjersing *et al.* 2010). However, standardization of a tool does not mean that it is good enough for use in a different culture or context or another time period where cultural contexts of society change (Malhotra *et al.* 1996). Having said this, there is no universal standardized procedure for the adaptation of scales to different cultures. A translation monitoring form was developed by researchers for a methodical preparation of instruments for transcultural use (Van Ommeren *et al.* 1999). A simple linguistic translation of scales does not yield cultural standardization and sometimes fails to reflect what the instrument is supposed to measure leading to imprecise and biased results. On the contrary, studies which have conducted comprehensive linguistic translation may not ensure construct validity and reliability (Beaton *et al.* 2000).

In a couple of Indian studies on the validation of Internalized stigma scale (ISMI) of Mental illness in Hindi and Malayalam languages, the authors found that translation of the scale was necessary due to difficulty in understanding the scales by lay persons and that some of the items of original scales were less applicable and hence, these items had to be removed considering the family, native cultural beliefs and religion (James *et al.* 2016; Singh *et al.* 2016b). In addition, Kumari and others reported that the affiliate stigma scale had no culturally sensitive items and hence, the psychometric validation was done without cultural adaptation (Kumari *et al.* 2022).

Cross-cultural adaptation of scales will help in the adaptation of questionnaires based on cognitive and emotional attachments of people to their cultures after the questionnaires have been translated from one language to another (Chu and Zhu 2023). Testing this process is laborious and each step is clearly defined in the cultural adaptation of scales (Arafat *et al.* 2016). In a way, cross-cultural adaptation can be viewed as both a process and a result, which is inevitably influenced by local cultural factors and individual capabilities.

Cross-cultural adaptation of scales reduces bias while assessing the attitude of people toward mental health because people's attitude and beliefs about mental illness differs in different parts of the world which can influence stigma towards mental illness (Abolfotouh *et al.* 2019). In this study, we culturally adapted six stigma scales into Kannada by following the translation steps of comprehensibility, acceptability, relevance and completeness as delineated by Van Ommeren and others. (Van Ommeren *et al.* 1999). Kannada is a Dravidian language spoken by the people of Karnataka state, located in the south of India. It is the eighth largest state of India with a population of 61,130,704 inhabitants ('Census' 2011), 31 districts and 240 talukas. Kannada was recognized as the official language of Karnataka in 1963 (Sharada 2002). In this study we aim to describe the cross-cultural adaptation process of six stigma assessment scales in Kannada and to utilize the qualitative findings from cognitive interviews to validate the scales to the native culture.

## Methodology

The six stigma scales used in this study were developed by researchers of The International Study of Discrimination and Stigma Outcomes (INDIGO) network (https://www.indigo-group.org/new-guide-to-scales/). The Indigo Partnership among low- and middle-income countries (LMICs) (China, Ethiopia, India, Nepal and Tunisia) carried out research to strengthen the understanding of mechanisms of stigma processes and reduce stigma and discrimination against people with mental health conditions in LMICs; and to establish a strong collaborative research consortium through the conduct of the program. Specifically, the Indigo Partnership involves developing and pilot testing anti-stigma interventions at the community, primary care and mental health specialist care levels, with a systematic approach to cultural and contextual adaptation across the sites. This work also involves transcultural translation and adaptation of stigma and discrimination measurement tools (Gronholm *et al.* 2023). As a part of this research study, we chose a rural community at Ramnagagara as the study site, as we had good networking with the community. This project had multiple interventions, two of which were conducted in the rural community and one in the urban community. Ramanagara is one of the 31 districts in Karnataka. This district is near to Bengaluru rural division.

A set of guidelines was prepared for use across sites of the INDIGO partnership using the systematic use of strategies advocated by Van Ommeren in their translation monitoring form (Van Ommeren *et al.* 1999). Along with the given guidelines we consulted several other guidelines for an effective and feasible method of adaptation of the scales (Younan *et al.* 2019; Beaton *et al.* 2000). We followed a five-step process to adapt six scales from English (original) to Kannada (target) language (Table 1). Two research team members (R1 and R2) were involved in the forward and backward translations, cognitive interviews and in systematically updating the scales at each stage. Three mental health experts (Ex1, Ex2, Ex3) reviewed all the translated versions of the scales and two other research team members (SL and AC) reviewed them for cultural equivalence and problems identified through the cognitive interviews. However, none of the items from the original scales were deleted nor any new items were added.

### Instruments

*Original scales can be accessed from INDIGO Partnership research program (https://www.indigo-group.org/new-guide-to-scales/).*

### Mental Health Knowledge Schedule (MAKS)

A 12-item scale that was developed and validated to measure participants' stigma-related knowledge or knowledge items that are important for stigma/ stigma reduction. This scale has two parts where part A comprises six items measuring stigma-related mental health knowledge and part B comprises six items that enquire about the classification of various conditions as mental illnesses (Evans-Lacko *et al.* 2010).

### The Reported and Intended Behavior Scale (RIBS)

An 8-item instrument, this scale was developed and validated to assess participants' reported and intended behavioral discrimination against persons with mental illness across four different domains. The first part, item 1–4 assesses the prevalence of the behavior and the second part, item 5–8 assesses willingness to engage in the stated behavior (Evans-Lacko *et al.* 2011).

### Discrimination and stigma scale short version (DISCUS)

A shorter version of the Discrimination and Stigma scale was developed to improve the usability of the scale. The original scale comprised 32 items that measured experienced and anticipated discrimination reported by people with mental health problems. The current shortened version consists of 11 items that captures the experiences of discrimination concerning aspects of everyday life (Bakolis *et al.* 2019).

### Social Distance Scale (SDS)

A 12-item scale is an adaptation of the social distance measure that was developed to measure the desire for social distance, i.e., whether respondents want to interact with people with mental illness (Penn *et al.* 1994).

### Stigma Stress Scale (SSS)

This scale measures whether respondents with mental illness perceive that the stigma associated with mental illness is a stressor for them (Rüsch *et al.* 2009). One item measures perceived stigma-related harm, and another item measures perceived resources to cope with stigma. The resulting difference score (harm minus coping resources) indicates the level of stigma stress. For this study, we use a brief 2-item version of the original 8-item scale.

### Attitudes to addressing stigma (ASTA)

Attitudes of mental health professionals to working to address the impact of stigma on service users will be measured through the attitudes to address stigma and discrimination scale (ASTA). This scale was created by rewording an existing scale, the Short Alcohol and Alcohol Problems Perception Questionnaire (SAAPPQ) (Anderson and Clement 1987). By retaining the questions and changing the wording from working with people with alcohol problems to working to reduce stigma and discrimination, the ASTA retains the structure of the SAAPPQ.

### Step 1: Forward translation

All the scales were translated from the original English version to the target language independently by two members of the research team. We followed a multiple-translation design. R1 and R2 independently translated all six scales from the original language to the target language. Both versions were then collated.

### Step 2: Expert review and synthesis

The translated versions were reviewed by experts for the appropriateness of the translation. Experts (Exp1, Exp2, Exp3) reviewed for readability, complexity and syntax and provided alternatives along with rationale for whichever item they felt was inappropriate. All the suggestions, comments and changes made by the experts were documented. A synthesis process in which different suggestions were collated and suitable changes were made to the items across the scales was carried out.

### Step 3: Reviewing for cultural equivalence

We developed an equivalence framework based on the work of Flaherty and colleagues who suggested five types of equivalence– Semantic, Content, Technical, Conceptual and Criterion (Evans-Lacko *et al.* 2010; Van Ommeren *et al.* 1999; Flaherty *et al.* 1988) (see supplementary for framework matrix of the modified equivalence criteria). Each item from all of the forward-translated tools was reviewed based on this framework by the research team members who were not part of forward translation. Any item that was not appropriate according to the criteria, was flagged and the second step was repeated for those items.

### Step 4: Back Translation and synthesis

All the reviewed scales were back-translated to the original language by members of the research team. R1 and R2 independently back-translated all the scales. The back-translated version and original version were compared and contrasted by SL and AC, research team members who were not involved in the translation process. Any discrepancy found was flagged and steps two and three were repeated for those items.

### Step 5: Cognitive interview

Cognitive interviews are a qualitative method to identify and analyze sources of response error in psychological scales (Willis and Artino 2013). In the process, the researcher tries to understand whether the conceptualization of constructs and comprehension of the items in the scale are perceived as intended by the participants. We adapted a cognitive probe framework adapted from Willis and Artino (2013) where probes guide the participant to think in a specific manner. We used proactive probes (systematically thought-out probes before the interview) and reactive probes (non-standardized probes asked often in response to participant behavior/responses). A cognitive debriefing form was used to record any sort of logistical and structural biases observed during the interview (see supplementary for the debriefing form). The forms were filled by the researchers immediately after each interview and helped in briefly reviewing the interviews and identifying items that require additional information from the participants.

Using the INDIGO partnership cognitive interview guide (see supplementary "cultural equivalence criteria" and "cognitive interview probes") R1 and R2 carried out cognitive interviews. Both of them conducted two pilot interviews each that were reviewed by two other research team members, SL and AC, to identify any potential interviewer biases. Around 35 people were approached for the cognitive interviews, of which eight service users could not

**Table 1.** shows the examples of each step of the cultural adaptation for six scales

| Sl. No. | Steps involved in test adaptation | Example | | Process |
| | | Source language | Target language | |
|---|---|---|---|---|
| 1 | Forward translation | I want to help to reduce stigma. (ASTA) | *kalankavannu kadime madalu naanu sahaya madalu bayasuttene* (I want to help reduce stigma) | Item independently translated by research team members. |
| 2 | Expert review and synthesis | …willing (RIBS, SDS) | *Siddaviddene* (ready) (R2 and Ex1), *tayariddene* (ready) (Ex3), *sammatisuttene* (willing) (Ex3), o*pputtene* (agree) (Ex2, Ex3. R1) | Multiple alternatives were given by experts when there was a discrepancy in translation, difficulty in comprehension and inappropriate to the target community. A consensus was then made to finalize the appropriate term. |
| 3 | Cultural equivalence | Have you been treated unfairly in dating or intimate relationships? (DISCUS) | *prema sambanda/nikhata sambandhagalalli nimmannu endadaru sariyillada reetiyalli nadesikollalagideya?* (Have you ever been treated unfairly in your love/intimate relationship) | Each item was reviewed using cultural equivalence criteria. Certain items/constructs changed based on cultural equivalence. |
| 4 | Back Translation and Synthesis | I am inclined to feel I failed to reduce stigma (ASTA) | I feel like I'm failing to reduce the stigma. | Original and back translations were compared and contrasted to check the equivalence between translated and original items. |
| 5 | Cognitive Interviews | I am able to rise up and meet the demands posed by prejudice against people with mental illness (SSS) | Are you able to cope with the discrimination about mental illness? | An in-depth interview with target participants was carried out to assess the suitability |

complete the interviews. The service users who dropped out were either symptomatic or did not understand the instructions despite repeated attempts. All the interviews were audio recorded with the consent of participants. The recruitment and setting of interviews varied for participants (see Table 3). All the responses were manually coded using the categories of INDIGO cultural equivalence framework. The four pilot interviews were also included in the final analysis.

## Experts and researchers background

All the research team members and experts from Bengaluru were bilingual mental health professionals who had knowledge of scale construction and the culture of target groups. The expert team comprised two psychiatrists and one epidemiologist from the National Institute of Mental Health and Neurosciences (NIMHANS). All of them had more than ten years of experience in the mental health field. The expert team was not part of the ongoing study. The research team comprised a psychiatrist, a psychologist and two psychiatric social workers from NIM-HANS. The cumulative team experience in the mental health field is around 35 years.

## Participants

Participants for the cognitive interviews were chosen using purposive sampling. The participants details are provided in Table 3.

## Data analysis

We manually coded the responses from the cognitive interviews using the five labels of cultural equivalence (comprehensibility, acceptability, relevance, response set and completeness) to compare with the original tools. The coding framework allowed to capture the general cognitive model of question-responses like unknown terms, ambiguous concepts, recall difficulty, estimation difficulty

and incomplete response options. The verbal responses were coded by R1 and R2. The researchers identified all the problems associated with the cognitive processes and extracted them in a tabular format. Verbatim quotes to support the identification of the problem were also noted. The other research team members along with R1 and R2 reviewed the problems and wherever a consensus could not be made on the adjustments to the questionnaire, the items were flagged to the original authors of the tools. After careful consideration, all original authors agreed to rephrase and modify the questions based on the Indian cultural context without changing the original meaning of the question. The identification of problems and adjustment of the items across the six scales is presented in Table 2. The debriefing forms were reviewed after each interview to check the mediating factors in the responses and identify items that needed further clarification from other participants. Although most of the interviews were conducted in-person without the presence of a third person and their influence, two interviews with community health workers and two interviews with community members could have had skewed responses as they occurred in the presence of other people who visited health care and neighbors in the community.

## Results

### Forward translation

R1 and R2 identified differences in the translations with names of the disorders (Schizophrenia was translated as *split mind* and *Capricious*; MAKS), name of the scale ('Reported' was translated as reported and narrated; RIBS), other words (Employment was translated as *job* and *work*; SDS) and syntax ('Have you been avoided or shunned by people who know that you have a mental health problem' was translated as '*Have you ever had people who know about your mental health problem kept a distance or turned away from you?*'; DISCUS). All these differences were then discussed, and a consensus was arrived at. Differences that could not be agreed upon were flagged for experts' review.

**Table 2.** Back translation of Discuss scale. (Illustrates the process of translation and changes in the concepts; see Discussion.)

| DISCUS scale items | | | | |
|---|---|---|---|---|
| Original item | Translated item | Back translated item | Finalized item | Finalized back translated item |
| Keeping the job | *kelasa maduvaga* | Doing the work | *kelasa ulisikolluvaga* | Keeping the work |
| Different levels of privacy | *nimma goupyate* | Your privacy | *aneka sandarbagalalli nimmannu sariyillada reethiyalli* | Treated unjustly in different situations |
| Shunned | *kantappisudvu* | Avoiding | *Kadeganisu* | Shun |

**Table 3.** Details of the participants recruited for cognitive interviews

| Participants | Characteristics | | | | |
|---|---|---|---|---|---|
| | N | Female | Age range (median) | Scales administered | Recruitment and Interview setting |
| Community Members | 6 | 4 | 27–48 (34) | MAKS, RIBS, SDS | Door to door survey, interview conducted at participants house |
| Community Health Workers | 3 | 3 | 26–52 (30) | MAKS, RIBS, SDS | Community Health Centre |
| Service Providers | 5 | 4 | 28–46 (32) | MAKS, RIBS, SDS | Community Health Centre |
| Service Users | 6 | 4 | 25–55 (38) | DISCUS, SSS | NIMHANS |
| Mental Health professionals | 7 | 7 | 28–42 (32) | ASTA | NIMHANS |

### Experts review and synthesis

The experts identified discrepancies in translation ('medications can be an effective treatment' was translated as *medication is effective;* MAKS), identified items that were difficult to comprehend (the translation for the items 'prejudice', and 'not realistic' were flagged as difficult to comprehend; SSS, ASTA), and identified items that appeared inappropriate to the target population ('community forest' appeared as an inappropriate construct to the target population; SDS). Experts suggested alternatives for all these items. All minor suggestions like inflectional endings and changes in the arrangement of words to ease comprehension were incorporated. Major changes and alternative suggestions were collated and the research team members (SL and AC) who were not involved in the translation, reviewed and reached a consensus. See the supplementary file ('Illustration of the process of expert review and synthesis') for scale-wise examples.

### Cultural equivalence

After expert review, each item was subjected to the cultural equivalence criteria. R1 and R2 independently reviewed the items according to the set criteria. All the items across the six scales were coded as technically equivalent with a few minor changes. Items that concerned any of the criteria were collated and put across to the expert's team and other research team members. See the supplementary file ('Illustration of the process of cultural equivalence review') for scale-wise examples.

### Back translation

Three items in the back-translated version of the DISCUS scale were found to be different from the original version. None of the other scales had any significant discrepancies.

### Cognitive interview

Table 3 shows the details of the participants recruited for the cognitive interviews. Community members and service users were aged 25–55 years and either had no formal education or had a range of low and high education, community health workers and service providers were mostly educated women. Service users have recovered individuals with varied diagnoses: schizophrenia, depression, alcohol dependence syndrome, anxiety, adjustment disorder and bipolar disorder.

Table 4 shows the coded responses from the cognitive interviews under the five domains of the cultural equivalence criteria where C stands for comprehensibility, A stands for acceptability, R for relevance and E for technical equivalence C for completeness. The underlying number under each domain indicates the number of participants who expressed difficulties under these domains when the said Precognitive Interview items were given to them for each scale. Verbatim quotes to support the identification of the problems and the adjusted questionnaire after reviewing the item and reaching the expert consensus are also shown in the table.

### Discussion

We describe the process of transcultural translation and adaptation of six stigma scales MAKS, RIBS, SSS, SDS, DISCUS and ASTA from English to the local language Kannada. A total of 27 participants were interviewed and the scales were adapted through the various stages of the adaptation process. Previous researchers who translated some of the stigma scales in different Indian languages have shown that the translation and expert review was the crux of their adaptation process in addition to conventional psychometric validation (Kumari *et al.* 2020; Mukherjee *et al.* 2017).

A few South Asian studies have culturally adapted stigma scales and used conventional forward and backward translation processes along with psychometric validation for the adaptation process

**Table 4.** Results from cognitive interview analysis

| Scale | Pre CI item | C | A | R | E | C | Quotes (translated to English) | Adjusted item |
|-------|-------------|---|---|---|---|---|--------------------------------|---------------|
| MAKS | *manasika arogyada samasye…* (Mental health problem) | 3 | | | | | "It feels like a word from a text book which we generally don't use in our daily conversations" | *Manasika samasye…* (Mental problem) |
| | *manasika samasye inda balaluttiruva anekaru aarogya seve odagisuva tagnyara sahaya padeyalu hoguttare* (Most people with mental health problems go to professional health care providers for help) | 4 | | | | | "Can you repeat once more? This is too lengthy and complex to understand…" | *Manasika samasye iruva anekaru sahayakkagi tagnyara bali hoguttare.* (Most people with mental health problems go to professionals for help) |
| | *khinnate* (Depression) | 5 | | | | | "Does this mean irritated?" | |
| | *icchita vikalate* (Schizophrenia) | 5 | | | | | "Because of physical disability people might develop mental illness" | |
| | *unmada khinnate* (Bipolar) | 5 | | | | | "*unmada* means anger, right? I think anger to an extent is normal… | |
| RIBS | *manasika arogyada samasye…* (Mental health problem) | 3 | | | | | "It feels like a word from a textbook which we generally don't use in our daily conversations" | *Manasika samasye…* (Mental problem) |
| | \*\**bhavishyadalli nanu manasika aarogya samasye iruvavara jote vasisalu siddaviddene* (In the future, I am willing to live with people having mental health problems) *bhavishyadalli nanu manasika aarogya samasye iruvavara hattira vasisalu siddaviddene* (In the future, I am willing to live near those who have a mental health problem) | 4 | | | | | "The fifth question you asked was also the same I think…." | *bhavishyadalli nanu manasika samasye iruvavara jote vasisalu siddaviddene* (In the future, I am willing to live with people having mental health problems) *bhavishyadalli nanu manasika samasye iruvavara mane hattira vasisalu siddaviddene* (In the future, I am willing to live near to the house of those who have a mental health problem) |
| SDS | *Manasika rogaviruva yarindadaru vastugalannu kollalu neevu yeshtu siddariddeeri?* (How willing are you to buy things from someone with a mental illness?) | 2 | | | | | "…I think you can use the word '*khareedi*' as that will be easier to understand" | *Manasika rogaviruva yarindadaru vastugalannu khareedisalu neevu yeshtu siddariddeeri?* (How willing are you to buy things from someone with a mental illness?) |
| | \**Manasika rogaviruva yaradaru nimma athava nimma parichayada makkalannu aaraike madalu neevu yeshtu siddariddeeri?* (How willing are you to let your children or children of your acquaintance, to be taken care by someone with a mental illness?) | 3 | | | | | "…I do take care of children…. They also need care, right?" | *nimma athava nimma parichayada makkalannu yaaradaru manasika rogaviruvavaru aaraike madalu neevu yeshtu siddariddeeri?* (How willing are you to let your children or children of your acquaintance, to be taken care by someone with a mental illness?) |
| | *Manasika rogaviruva yarigadaru aarogya seveyannu odagisalu neevu yeshtu siddariddeeri?* (How willing are you to provide health care for someone with mental illness?) | 2 | | | | | "…what is *odagisalu*? Does it mean to 'do' something?……, *needalu* would be easy to understand than the word *odagisalu*" | *Manasika rogaviruva yarigadaru aarogya seveyannu needalu neevu yeshtu siddariddeeri?* (How willing are you to provide health care for someone with mental illness?) |
| | *Manasika rogaviruva yaradaru nimma sambandita vyaktiyondige maduveyaagalu neevu yeshtu siddariddeeri?* (How willing are you to have someone with mental illness marry someone related to you?) | | 5 | | | | "…If they are marrying by their will then it is fine…. If it was my relative, I would ask them to rethink/reconsider" | *Manasika rogaviruva yaradaru nimma sambandikarondige maduveyaagalu neevu yeshtu siddariddeeri?* (How willing are you to have someone with mental illness marry a relative of yours?) |
| SSS | *manasika khaliye hondiruva janara virudda iruva kalanka mathu bedhabhava nanna jeevanada vividha kshetragalalli (sneha-sambanda, maduve, kelasa, shikshana) tondareyaagide* (Stigma and discrimination against people with mental illness will affect | 6 | | | 6 | 6 | "…I didn't understand! … I have to answer, is it?" | *manasika khayile bagge iruva tappu kalpane, mathu bedhabaavadinda (aadikollodu, heeyalisodu, kettadagi nadesikollodu) nimma jeevanadalli (sneha-sambanda, maduve, kelasa, shikshana) tondare aagabahude?* (Will your life (friendships, marriage, |

**Table 4.** (*Continued*)

| Scale | Pre CI item | C | A | R | E | C | Quotes (translated to English) | Adjusted item |
|---|---|---|---|---|---|---|---|---|
| | several areas of my life (friendship, marriage, work, education)) | | | | | | | work, education) be affected by the misconception and discrimination about the mental illness (teasing, sneering, putting you down, treating you badly)?) |
| | *manasika khayile hondiruva janara virudda iruva kalanka mathu bedhabhavavannu nibhayisalu nanu shaktanagiddene* (I am able to cope with the stigma and discrimination against people with mental illness) | 6 | | 6 | 6 | | | *Manasika khayileya bagge iruva tappu kalpane mathu bedhabhavavannu nibhayisalu nimage sadhyave?* (Are you able to cope with the misconception and discrimination about mental illness?) |
| DISCUS | *vasatiyanni padeyuvalli nimmannu endadaru sariyillada reetiyalli nadesikollalagideya?* (Have you ever been treated unjustly while searching for a house?) | 4 | | | | | "… *vasati* means? … is it like able to earn and have a livelihood?" | *manasika samasyeyindagi mane hudukalu (athava yara maneyalladaru ulidukolluvaga) nimmannu yendadaru sariyillada reetiyalli nadesikollalagideye?* (Because of mental illness have you been treated unfairly while searching for a house or to stay at someone's house?) |
| | *nimma vayaktika surakshate mathu bhadrateyalli endadaru nimmannu sariyillada reetiyalli nadesikollalagideya? (moukhika nindane, daihika kirukula, halle bagge keli)* (Have you ever been treated unjustly in your personal safety and security? (Ask about verbal abuse, physical abuse, assault)) | 5 | | | | | "…which means? … I can't understand…" | *nimma vayaktika surakshate mathu bhadrateyalli endadaru nimmannu sariyillada reetiyalli nadesikollalagideya? (moukhika nindane - baiyuvudu, daihika kirukula - hodeyodu, halle–kirukula kododu bagge keli)* (Have you ever been treated unjustly in your personal safety and security? (Ask about verbal abuse - scolding, physical abuse - beating, assault–persecution)) |
| | *prema sambanda/nikata sambandagalalli nimmannu endaru sariyillada reetiyalli nadesikollalagideya?* (Have you ever been treated unjustly in love relationships or intimate relationships?) | | 5 | | | | "… I know love and relationships are wrong. I have left it now…" | Manasika samasye indagi maduve aguva sandarbadalli(gandu/hennu hudukuvaga) athava dampathya jeevanadalli nimmannu endaru sariyallada reethiyali nadesikollalagideye? Because of mental health problems, have you ever been treated unjustly while searching for marriage alliance or marriage life? |
| | *aneka sandarbhagalalli nimma khasagitanavannu (goupyate) kapadikolluvalli endadaru nimmannu sariyillada reetiyalli nadesikollalagideye? (aaspatreyalli mattu samudayadalli, khasagi patragalu athava dooravani karegalu, vaidyakeeya daakhalegala bagge keli)* In different situations, have you ever been treated unjustly about maintaining your privacy? (In hospital and in the community, *e.g.*, asking about private letters or telephone calls, medical records) | | | | 6 | | "… I don't keep anything for myself… I just openly tell everything to (my spouse) … what is there to hide? Everyone knows it…" | Manasika samsyeyindagi hanakasu vyavaharadalli nimmannu endaru sariyallada reethiyalli nadesikollalagideye?(aasthi hanchike vethana tharathamyada bagge keli) Because of mental health problems, have you ever been treated unjustly in your finances? (Ask about property distribution, salary norms, activities) |
| ASTA | *manasika khayile iruva janarondige kelasa maduvaga avara viruddada bedhabhavada bagge nanage iruva tiluvalikeyannnu upayogisabahudendu nanu bhavisuthene* (I think I can use my understanding of the discrimination against people with mental illness when working with them) | 6 | | | | 4 | "When my colleagues share anything about their mental health problems, I try to be open and helpful as much as I can…" | *Naanu rogigalondige kelasa maduvaaga, manasika khayile iruva janara viruddada bhedabhavada bagge nanagiruva tiluvalikeyannu upayogisaballe yendu bhavisuthene* (I think I make use of my understanding of the discrimination against people with mental illness when I am working with patients) |
| | *manasika khayile iruva janarige avara sthithiyannu itararige tilisi heluvudara bagge sooktavada salaheyannu needaballenendu nanu bhaavisuthene* (I think I can give appropriate advice to people with mental illness on explaining their condition to others) | 6 | | | | 3 | "I feel the sentence is too lengthy… I can give advice when they ask for it…" | *nanu nanna rogigalige avara sthithiyannu itararige tilisi helalu sookta salaheyannu needaballenendu bhaavisuthene* (I think I can give appropriate advice to my patients to explain their condition to others) |

(*Continued*)

**Table 4.** (Continued)

| Scale | Pre CI item | C | A | R | E | C | Quotes (translated to English) | Adjusted item |
|---|---|---|---|---|---|---|---|---|
| | *manasika khayile iruva janaru tamma viruddada bhedabhavavannu kadime madalu nanna sahaayavannu padeyalu icchisuthare endu nanu bhavisuthene* (I think that people with mental illness want my help in reducing the discrimination against them.) | | | | | 4 | "If they know me then they would ask for help…." | *rogigalu manasika khayileyindagi tamma viruddada bhedabhavavannu kadime madalu nanna sahaayavannu padeyalu icchisuthare endu nanu bhavisuthene* (I think, patients would want my help to reduce discrimination against them because of mental illness) |
| | …. I have failed to reduce the discrimination<br>…. It is impossible to try to reduce the discrimination | | | | 4 | 5 | "I can say it is difficult or I could have put more efforts, but thinking I have failed completely is unreasonable…" | Manasika kayile hondiruva janara viruddhada beda bhavavannu kadime maduvalli nanu hecchu prayathnisilla embuva bhavane moodutthave<br>I feeling that I have not put much effort into reducing discrimination against people with mental illness<br>Manasika kayile hondiruva janara viruddhada beda bhavavannu kadime madalu prayathinisuvudu sulabahavalla.<br>It is not easy to try to reduce discrimination against people with mental illness. |
| # MAKS, SDS, ASTA and RIBS | *sammatisuthene–assamatisuthene* (Agree–disagree)<br>Balavagi Opputthene (Strongly agree) | 5 | | | | | "hu opkotini… sammati ide … idu opkolovantade alwa …" (yes agree, … It's agreeable right. The english translation?) "Balavagi" was perceived by the participants as forcefully instead of strongly so it was changed to "kanditavagi" | *opputhene–oppuvudilla* (Agree–disagree)<br>Kandithavagi Opputhene (Strongly agree) |

\*\*Both the items were perceived to be the same. Added an additional word to make the difference explicit.
\*Changed the sentence structure and order of words.
#The changes were suggested for words of the Likert rating scale.

(Arafat *et al.* 2022; Gupta *et al.* 2023) Our study followed a systematic adaptation process of using cognitive interview techniques in addition to the above-mentioned processes to validate the content by service providers and service users. Although a few studies have considered expert review and face validation as a pretesting method of item appraisal (Baba *et al.* 2021; Dalky 2012; Wu *et al.* 2020), expert opinions may not be accepted without discretion and are not likely to reveal measurement errors (Jia *et al.* 2022; Ryan *et al.* 2012). Cognitive interviewing as a method of pretesting, addressed participants' comprehension of the items and provided an insight into the target culture and generated evidence that resulted in reframing/change in items and response sets (Beatty and Willis 2007; Miller *et al.* 2014; Srinivasan *et al.* 2021).

### Mental health knowledge schedule (MAKS)

In one of the statements, the participants felt that the questionnaire was complicated and lengthy so, alternate suggestions were given to simplify the statements from 'Mental health problem' to 'mental problem' for ease of understanding. Choi and Pak 2005, in the catalog of biases, mention that complex and lengthy terms are better avoided in questionnaires as they can confuse the subject and lead to incorrect answers. Some of the native synonyms of medical diagnosis were not easy to understand for the participants from the community. Terms such as 'depression' (was interpreted as irritation), 'schizophrenia' (interpreted as physical disability), 'bipolar' (interpreted as anger); as a result, both colloquial and

English words were retained in the scale for better understanding since these medical terminologies were borrowed words (Rao, 2018).

### The reported and intended behavior scale (RIBS)

Certain words had to be introduced for one of the items, (in the future, I am willing to live near those who have a mental health problem), we introduced the word 'live near the house of those……', to make the question more specific and clearer and to avoid duplication of responses (Madson 2005; Wright *et al.* 1997). One other item in the scale carried similar phrases (in the future, I would be willing to live with someone with a mental health problem) (refer Items 5 and 7 RIBS on Indigo partnership website). In the collectivistic Indian society, the joint family system consisting of adults living with parents and inheriting the ancestral property is common, and indeed more so in many parts of rural India as well (Chadda and Deb 2013; Mullatti 1995). Thus, searching for houses is less prevalent in Indian rural areas as compared to urban areas where migrant population/adults from other towns launch out from their families (launching out from family is the stage of individuation in the family cycle where an individual leaves the family of origin) and reside in rental homes or buys a new home to live as a nuclear family (Datta 2011; Holmström 1970). Thus, the concept of housing is probably more suited to the individualistic Western culture where the individuation of adults is observed earlier in life. So, we modified the question to experiences of 'staying

in a relatives house' or 'renting a house' to assess the stigma and discrimination associated with housing.

### Attitudes to addressing stigma (ASTA)

It is not unusual for experts to suggest changes in words or phrases from a stigmatizing to a less stigmatizing version. However such modifications may lead to response bias when administered to the subjects (Madson 2005). For example, in one of the questions here, we found differences in perspectives between experts and respondents during the cognitive interview: 'I think I make use of my understanding of the discrimination against people with mental illness when I am working with patients'; the expert recommended to remove the word 'patient' considering a possible negative connotation attached to it. However, during the cognitive interview, responses were skewed when the question was framed in the generic form, so we retained the word 'patient' in the final version as it existed in the original. Retaining this word was not stigmatizing since the scale was administered to mental health professionals and the word 'patients' gave more clarity to them about the population the question is referring to, rather than the word 'people'. This suggests that sometimes clarity and specificity of the question determine the responses (Krosnick and Presser 2010). We also observed that using negative affirmative statements/concluding remarks resulted in ceiling-level responses among the subjects. For example, 'I feel that I have failed to reduce the discrimination against people with mental illness' (item 4 of ASTA scale on Indigo partnership website), was revised to 'I am feeling that I have not put much effort into reducing discrimination against people with mental illnesses. 'It is impossible to try to reduce the discrimination against people with mental illness' (item 6, of ASTA scale in Indigo partnership website) was revised to, 'It is not easy to try to reduce discrimination against people with mental illness.' Hence, constructive statements like 'did not put much effort' in place of 'failed', 'It is not easy' instead of 'impossible' were used. Statement polarity in self-report measures can significantly affect the rating, often resulting in biased and ceiling responses (Chyung et al. 2020; Kamoen et al. 2013). The authors also opine that respondents are more likely to disagree with negative questions than to agree with positive questions or to choose the positive side of the bipolar scale. Thus, items like, 'It is impossible…, I have failed…., I have not achieved…,' were perceived to be negatively associated with participants' skills which resulted in the ceiling responses. Based on the responses of the participants the statements were modified to moderate polarity.

### Social distance scale (SDS)

In the Indian context, the service user's prospects of their marriage being affected due to stigma associated with mental illness is a well-researched finding (Koschorke et al. 2017; Loganathan and Murthy 2008; Raguram et al. 2004; Srivastava 2013; Weiss et al. 2001). The item, 'How willing are you to have someone with mental illness marry someone related to you' did not capture the intended responses due to ambiguity on the relationship in the phrase 'someone related'. When the relationship status was changed to indicate a 'first degree relative', the responses varied, and participants indicated their unwillingness for the item (refer to item number 3, SDS in Indigo partnership website). In a collectivistic Indian society, families prefer to babysit their children with the support of their own family members or relatives, and the concept of babysitting is not as prevalent as in the West (Medora 2007). Therefore, the experiences of stigma related to babysitting had skewed responses (responses at one extreme–strongly disagree). The wordings of the item on babysitting were changed to explicitly provide a hypothetical situation for the participants, 'How willing are you to let your children or children of your acquaintance, to be taken care by someone with a mental illness?'

### Short version of DISC (DISCUS)

The experts gave empirical opinions that helped conceptualize a few expressions across the scales like 'treated unfair' or identifying the non-stigmatizing yet informal and common words for addressing people with mental illness. Thus, equivalent constructs for community forests were replaced by 'self-help groups', 'small business' (chit funds),/local governance and for 'treated unfair' words like 'treated unjustly'/'treated in an inappropriate' way were used (refer DISCUSS item 1 to 9 in the Indigo partnership website).

The concept of privacy with respect to phone calls, medical records, private letters and criminal records was changed to 'unjustly treatment of person with mental illness in finance, property distribution, salary norms and activities. The respondents stated that it does not make any difference to them as they shared everything with their partners. The concept of privacy is familial than individual and can be said to be virtually absent in the Indian rural context. (Chadda and Deb 2013; Manzar and Chaturvedi 2017). Other terms that were not culturally relevant to the rural Indian context like 'dating' or 'intimate relationship' were changed into marriage alliance and marital life, respectively. The participants of the cognitive interviews felt uncomfortable and hesitant, as direct questioning on dating and intimate relationships was generally not accepted in the rural Indian cultural context (Manjistha et al. 2013). During the interviews with the participants who were married or when the question was reiterated within the context of marital relationships, the intended responses were observed. Hence, the item was changed to the context of 'marital relationship'.

### Stigma stress scale (SSS)

Using technical vs. colloquial terms was one of the key contentions among the expert committee. For items like 'prejudice', the colloquial term was not appropriately capturing the original meaning and the technical word was not commonly understood by the layperson. It was decided to keep the language informal and simple, hence 'stigma and discrimination' was used instead of 'prejudice' There is no colloquial equivalent for the term 'prejudice' in Kannada. In addition, suitable scenarios were given to make the questions more relevant. For example, the first item of SSS mentions 'Prejudice against people with mental illness will affect many areas of my life' which was culturally adapted as 'Has your life (friendships, marriage, work, education) been affected by the misconception and discrimination about mental illness (teasing, sneering, putting you down, treating you badly)?' While developing the Hindi version of a different stigma scale called, Internalized Stigma Scale of Mental Illness, the author used simple colloquial words rather than technical words which was easy for lay people to understand (Singh et al. 2016a), emphasizing that scales with technical jargons may not be useful in eliciting the intended responses.

## Response set of all scales

In self-report measures, the statement format, wordings, response scales and anchors for responses can directly impact participants' answers (Chyung *et al.* 2018a, 2018b). In the SSS scale, the participants did not understand the self-reference when items were given as statements, e.g.,–*manasika khayile hondiruva janara virudda iruva kalanka mathu bedhabhava nanna jeevanada vividha kshetragalalli (sneha-sambanda, maduve, kelasa, shikshana)* Stigma and discrimination against people with mental illness will affect several areas of my life (friendship, marriage, work and education). Instead, when the items were presented as questions, participants were able to answer clearly without repeating the item or asking for elaborative explanations. Similarly, the response set in the original scale provided only the numbers without anchors which the participants found difficult to provide responses. The responses were then provided with suitable anchors. On the other hand, the anchors for the response set in MAKS, RIBS and SDS scales were changed based on the responses of the participants to make it easier to follow (refer supplementary response sets for all scales).

## Strengths and limitations

This study provides comprehensive details of the cross-cultural adaptation of six stigma scales for making an ecologically valid tool. This is one of the prototype studies from India that followed a rigorous and systematic process of adaptation of stigma scales using recognized guidelines. An attempt has been made to retain the originality of the questionnaires, yet, meeting with the cultural contexts. The study does have a few limitations. Some cognitive interviews were conducted in the presence of others which may have influenced the participants' responses. R1 and R2, the two research members of the team were involved in both the forward and back translation process as well as the cognitive interviews, though other research members were also involved in the latter two steps. This could have introduced confirmation bias. Also, Kannada is spoken in three dialects in the state: in the north, south and coastal Karnataka. Thus, regional variations may limit the wider use of this scale and may necessitate further changes in these parts of the state.

## Conclusion

For several years stigma researchers in India have either relied on Western instruments or semi-structured stigma scales in their studies. As a result, many of the Western concepts could not be well understood by participants, and were sometimes misunderstood as well, since stigma and discrimination are commonly experienced in other contexts such as caste, gender and poverty in the Indian sociocultural context. Several of the semi-structured stigma scales that were used earlier lack the rigorous standardization that is required for any scale to be used consistently and repeatedly. However, these semi-structured stigma scales also had a few qualitative questions on stigma in addition to items and responses to capture the rich variety of experiences that the scales alone may not be able to elicit. We hope that the cultural adaptation process described in this study is useful for other researchers wanting to decode and adapt these scales to other languages in India and possibly other low- and middle-income countries. With the translation and cultural adaptation of these six instruments to Kannada, it will now be available for stigma researchers to use them in the future. Although it is essential to have a rigorous adaptation, the psychometric properties are equally important to establish the validity and reliability of the tests.

**Open peer review.** To view the open peer review materials for this article, please visit http://doi.org/10.1017/gmh.2024.84.

**Supplementary material.** The supplementary material for this article can be found at http://doi.org/10.1017/gmh.2024.84.

**Acknowledgements.** We acknowledge our participants who took part in the cognitive interview and we also appreciate the efforts of the neutral reviewer (mental health professionals) who provided us excellent expert opinion. We acknowledge Rahul (RA1) who was involved in data collection and cognitive interview.

**Author contribution.** The first authors (HAA, GBM) equally contributed to conceptualizing the article and played a principle role in writing the complete draft. The corresponding author (SL) conceptualized the methodology and reviewed article for better quality. Other authors (GT, AVC, BK, NR, SW, SE, BE, CH) contributed equally in reviewing the article and modifying the article. All authors contributed to, read and approved the final manuscript.

**Financial support.** INDIGO partnership project, collaboration with King's College London, and Funded by Medical Research Council (MRC), United Kingdom (Grant reference: RE14404, MRC reference: MR R023697).

**Competing interest.** None declared.

**Ethics statement.** Approval from Institute ethics committee, Behavioral science (Division), National Institute of Mental Health and Neuro Sciences, Bengaluru. Ethics approval number (NIMHANS/EC (BEH.SC.DIV.)20[th] Meeting/2019 Dated: 23.09.2019).

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
