## [Reviewer Report]

Could you kindly let me know which new things you have added to this paper? I observed that you just added some previous mental health scale details.

---

## [Reviewer Report]

poorly written abstract, lack of methodology, and basic abstract format, writing is poor, sample size calculation is missing, study population and settings are missing, methodology not sound.

---

## [Reviewer Report]

This study reports the cultural adaptation process of stigma assessment scales among the Kannada speaking population. The study is methodically designed and implemented for the adaptation of stigma scales. Although, the following limitations have been observed in the study are highlighted below:

1. Authors may also report the rationale behind the selection of six stigma scales viz. MAKS, RIBS, SDS, SSS, DISCUS, and ASTA for translation and cultural adaptation into Kannada. The authors should also justify why the adaptation process is only implemented in the rural part of Karnataka.

2. Sentence 11-12: What could be the possible risk of biases in the context of the outcome measure of stigma reported in the meta-analytical studies conducted by Gaiha et al. 2020. Explain.

3. Page 8 Sentence 12: Authors have reported that although most of the interviews were free of external mediating factors. What were the mediating factors, specify.

4. Authors must present all original English items of MAKS, RIBS, SDS, SSS, DISCUS, ASTA, and report the equivalent Kannada translation, and further should discuss the comprehensibility, acceptability, relevance, and technical equivalence, and completeness for each item of said items in tabular form. Further, after the completion of the cultural adaptation of all the applicable items, revised items in Kannada should be also reported so that the actual revision of all the items of the six stigma scales can be reflected in the manuscript.

5. In the results section, modifications of words, sentences, and syntaxes are abruptly described in the manuscript. As a reader, it is difficult to comprehend in what context what changes have been made, which could provide a better understanding of the adaptation that has been done for each item of the six stigma scales.

---

## [Reviewer Report]

This paper provides a comprehensive and systematic adaptation of six stigma scales among Kannada speaking people in India. I have provided some comments for the authors’ consideration.

Background

1. The introduction can be shortened for brevity. For example, the statement from line 52-60 may not be necessary as it does not tie in well with the preceding sentence or the ones which follow.

2. It may be helpful to provide some context of the Kannada speaking people (or Kannada language), for readers who may be unfamiliar with the sociocultural structures of India.

3. Throughout the paper, reference is made to the INDIGO project. Although many readers in the field may be familiar with the project some may not. Adding this information will make the paper stand alone.

Methods

4. Consider rewording the second aim of the study for clarity. The ‘’and’ may be misplaced? Otherwise, some information is missing.

5. Researchers R1 and R2 conducted both forward and back translations which is not standard practice. This may have introduced confirmation bias.

6. For reproducibility, consider providing the full set of cognitive probe frameworks used.

Discussion

1. Discuss the limitations of having R1 and R2 conducting forward and back translations as well as the cognitive interviews

---

## [Reviewer Report]

This paper describes the cultural adaptation of six scales for the assessment of stigma in a particular state of South India.

As there is a clear need for instruments of this sort to be available in local languages, research of this sort is amply justified. The paper is well written and organized in a logical manner, and the translation process has been reported in detail, including examples for each step. Further details are available in the supplementary material for interested readers or specialists. In my opinion, the authors are to be commended for undertaking this work.

The following are aspects of the paper that would benefit from correction or clarification:

1. Title: The number of scales adapted (6) could be mentioned in the title. “Stigma assessment scales” on its own is indeterminate and gives the impression of a review paper rather than original research.

2. Abstract: The abstract should be structured in line with standard guidelines for original research. The sub-heading “Methods” can be used to describe the process of adaptation, and the measures of reliability and validity for the adapted instruments can be presented under “Findings” or “Results”.

3. Introduction:

a. As far as possible, avoid “etc.” in a scholarly work. The “demand” and “supply”-side barriers can be enumerated separately and succinctly.

b. More details of the current state of knowledge / research in this field could be provided. For example: What other Indian languages have stigma assessment tools been translated into? What were the challenges faced by the developers of these modified versions? How would Indian culture affect the validity of specific items / domains in existing tools, and thereby necessitate the modification of these items? It is true that this will be addressed in more depth in the Methods section, but some background information would be valuable.

c. The aims and objectives of the current work can be stated clearly (in 2-3 sentences) either at the end of the introduction, or under a separate sub-heading. They are alluded to at the end of the introduction in the current manuscript, but only in a general manner.

4. Methodology:

a. Why were these six particular scales chosen? Is there prior work on adapting any of these six scales into other Indian languages (e.g., Hindi)?

b. Did any items need to be deleted because they were irrelevant to the local context?

c. Likewise, were there any items that the experts (or other participants mentioned in Table 3) thought needed to be included, but did not exist in the original instruments?

d. The authors have carried out their work in a specified rural area of a larger state. Are there regional variations / dialects in Kannada that would limit the wider use of this tool, or necessitate further changes? (This can happen with other languages, both “Eastern” and “Western”.)

(Note that if the authors think that these issues can be clarified in the Results rather than in the Methods, I leave it to their discretion.)

5. Results:

Were any of the conventional measures of psychometric validity / reliability (e.g., measures of internal consistency, convergent validity between different tools) estimated for the six study instruments? If not, why? (If this is being addressed in a subsequent / future publication due to space constraints, the authors could simply state this.)

6. Discussion:

Apart from discussion of challenges related to each specific instrument, it would be worthwhile comparing the current work with prior research on adapting stigma assessment tools in the Indian (or South Asian) context. (If there were no comparable studies, this could be mentioned, but this is unlikely.) The Discussion should not only critically analyze the results, but place the current research in a broader context.

---

## [Reviewer Report]

The concerns raised in my earlier review report have been addressed by the authors in the revised manuscript.

I have no further major changes or corrections to suggest.